# Hybrid integrated ultra-low linewidth coil stabilized isolator-free widely tunable external cavity laser

David A. S. Heim ⓘ, Debapam Bose, Kaikai Liu ⓘ, Andrei Isichenko ⓘ & Daniel J. Blumenthal ⓘ ✉

Precision applications including quantum computing and sensing, mmWave/ RF generation, and metrology, demand widely tunable, ultra-low phase noise lasers. Today, these experiments employ table-scale systems with bulk-optics and isolators to achieve requisite noise, stability, and tunability. Photonic integration will enable scalable, reliable and portable solutions. Here we report a hybrid-integrated external cavity widely tunable laser stabilized to a 10 m-long integrated coil-resonator, achieving record-low 3 – 7 Hz fundamental linewidth across a 60 nm tuning range and 27 – 60 Hz integral linewidth with 1.8E-13 ADEV at 6.4 ms across 40 nm, delivering orders of magnitude frequency noise and integral linewidth reduction over state of the art. Stabilization is achieved without an optical isolator, leveraging resilience to optical feedback of 30 dB beyond that of commercial DFB lasers. The laser and reference cavity are fabricated in the same Si3N4 CMOS-compatible process, unlocking a path towards fully integrated visible to ShortWave-IR frequency-stabilized lasers.

Ultra-narrow linewidth widely tunable stabilized lasers are critical for a range of precision applications including optical atomic clocks[1,2], quantum computing[3–6], metrology[7,8], quantum and fiber sensing[9–11], and low phase noise mmWave and RF generation[12,13]. Of paramount importance to these applications is the phase noise as measured from low to high carrier offset frequencies and characterized in part by the instantaneous and integral linewidths (ILW). To achieve ultra-low linewidths, these laser systems utilize large mode volume lab-scale external cavity lasers, bulk-optic reference cavities[14,15] and optical isolation in the laser stabilization circuit. Integration using a wavelength-transparent platform will improve reliability, reduce size, weight, power, and cost, and enable scalability, portability, and systems-on-chip solutions across the visible to ShortWave-IR (SWIR). Yet, to date, integration of widely tunable, cavity-stabilized lasers in a CMOS foundry-compatible integrated platform has remained elusive. Low instantaneous linewidth integrated lasers include self-injection locked[16–19], stimulated Brillouin scattering (SBS)[20–22], external DBR (EDBR)[23], and external cavity lasers (ECLs)[24,25]. In particular, the

external cavity tunable laser (ECTL) design[26–28] is used due to its wide wavelength tuning range, low fundamental linewidth (FLW), and ability to stabilize the laser output to an external optical reference cavity for ILW reduction and carrier stabilization. Photonic integration of stabilized ECTLs is a critical step forward for robust solutions, operating from the visible to SWIR, to serve as stand-alone sources and as pumps for optical frequency combs, nonlinear wavelength conversion, and Brillouin lasers.

Silicon nitride ($Si_3N_4$) is an ideal integration platform[29] due to its low-propagation loss that extends from the visible to SWIR. The combination of ultra-high quality factor (Q) resonators[30,31], the ability to integrate gain media[32], and waveguide-compatible control and modulation[33–35] enable a wide range of systems-on-chip solutions. Silicon nitride photonics have been used to realize narrowly tunable low FLW lasers[36–39] and large mode-volume resonator reference cavities[40,41]. The high Q resonators enable increased intra-cavity photon lifetime and photon number needed to reduce the FLW, while large mode-volume laser and reference cavities decrease the

---

Department of Electrical and Computer Engineering, University of California Santa Barbara, Santa Barbara, CA, USA. ✉e-mail: danb@ucsb.edu

intrinsic thermorefractive noise (TRN)[42]. Additionally, a high laser resonator Q and other techniques can improve the resilience to optical feedback[43–47]. Hybrid integration of $Si_3N_4$ ECLs[23,25,26,28,36,37,39] is an effective way to combine the benefits of ultra-low loss waveguides and high Q resonators with III–V semiconductor gain materials at a wide range of wavelengths. However, achieving both low fundamental and low ILWs across widely tunable wavelength ranges, and without the need for optical isolation in a common integration platform that supports the ECTL, reference cavity, and other photonic components has not been achieved.

Here, we present a significant advance in chip-scale, widely tunable stabilized laser technology by demonstrating a coil stabilized isolator-free hybrid integrated $Si_3N_4$ ECTL with a 60 nm tuning range, FLW of 3–7 Hz across a full 60 nm tuning range and 27–60 Hz $1/\pi$ ILW across a 40 nm tuning range. These results represent a frequency noise reduction of over 6-orders of magnitude, almost 2-orders of magnitude reduction in ILW over the free-running linewidth, and 65 dB side mode suppression ratio (SMSR) across the tuning range. Linewidth reduction and carrier stabilization are achieved by directly locking the ECTL, without optical isolation, to a silicon nitride 10 m-long coil resonator reference. The 20 MHz free-spectral range (FSR) of the 10 m coil enables stabilization at almost any wavelength across the 60 nm tuning range. We also measure an Allan deviation (ADEV) of $1.8 \times 10^{-13}$ at 6.4 ms and 5.0 kHz/s drift. An accurate ILW measurement is achieved using an ultra-low expansion (ULE) cavity-stabilized frequency comb to heterodyne beatnote measure the laser noise down to 1 Hz frequency offset across a 40 nm tuning range. We demonstrate that the feedback resilience of the 3.5 million intrinsic-Q Vernier rings in the ECTL provides an inherent isolation of ~30 dB relative to a typical commercial III–V DFB laser. The ECTL and 10 m coil-reference cavity are fabricated in the same 80 nm thick silicon nitride low-loss platform. This common design and fabrication process, combined with the feedback resilience of the ECTL, enables a path towards full integration of widely tunable, narrow-linewidth, frequency-stabilized lasers at a

wide range of quantum, atomic, fiber communication, and other wavelengths from the visible through the SWIR.

## Results

### Coil-stabilized ECTL architecture and design

The coil-stabilized integrated tunable laser experiment, shown in Fig. 1a, consists of a widely tunable silicon nitride ECTL (left side of Fig. 1a) directly connected through an electrooptic modulator (EOM) for sideband locking to a 10 m coil resonator reference cavity (right side of Fig. 1a). The coil resonator output is tapped and photodetected and fed back to control the ECTL lasing frequency using a Pound–Drever–Hall (PDH) circuit. The ECTL and coil resonator are fabricated in the same 80 nm thick silicon nitride fabrication process and utilize the fundamental TE mode, where the waveguide width can be adjusted to optimize the tradeoff between bend radius and propagation losses due to sidewall scattering (see Supplementary Note 2 and 3 for more details on waveguide design).

The hybrid integrated ECTL (Fig. 1c) consists of a silicon nitride waveguide section with two actuatable high-Q intra-cavity ring resonators, an adjustable phase section, a tunable Sagnac loop mirror, and a gain section consisting of a butt-coupled InP reflective semiconductor optical amplifier (RSOA). The two high-Q rings, connected in an add-drop configuration, increase the intra-cavity photon lifetime and provide instantaneous linewidth reduction. Thermo-optically tuned actuators on the ring resonators enable wide single-mode tuning utilizing the Vernier effect, and the tunable Sagnac loop mirror serves as a tunable broadband cavity reflector. Additional fine lasing mode tuning is achieved using the thermally actuated phase section. Hybrid integration is achieved by edge coupling a Thorlabs single-angled facet C-band reflective SOA (SAF 1128C) to an 18 µm-wide angled waveguide fabricated at the input to the $Si_3N_4$ ECTL chip. The RSOA is mounted on a heat sink and wire bonded to an electronic circuit board. The input waveguide is designed to optimize the modal overlap between the gain chip and the $Si_3N_4$ PIC before tapering down

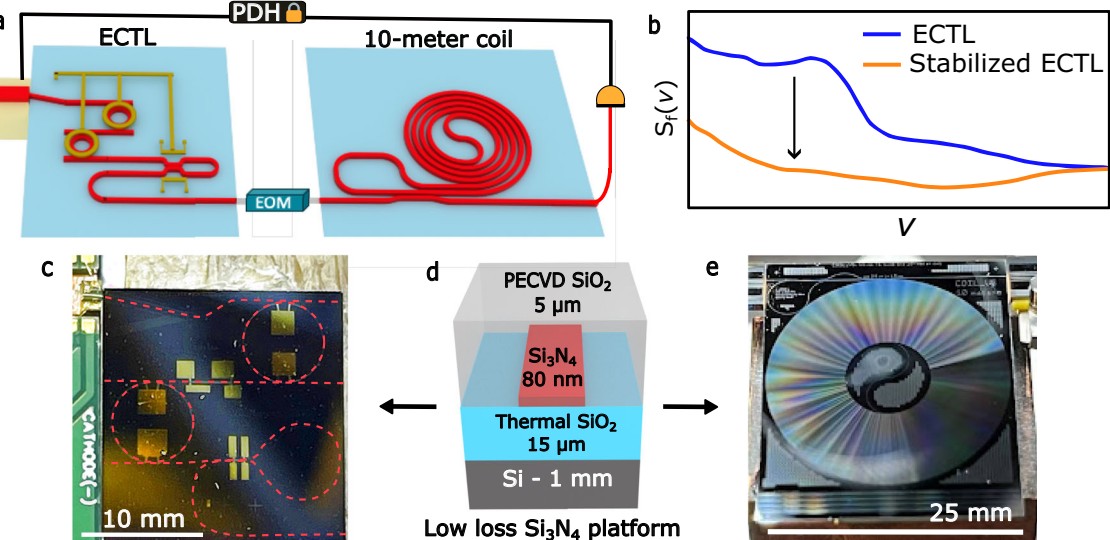

**Fig. 1 | Experimental setup of the integrated ECTL stabilized to an integrated coil reference cavity. a** Schematic of the experimental setup where the external cavity tunable laser (ECTL) is PDH-locked to an integrated 10 m coil reference cavity without an optical isolator. Hybrid integration of a reflective semiconductor optical amplifier (RSOA) provides a gain element. The high quality factor (Q) silicon nitride ($Si_3N_4$) rings serve as an external cavity and provide instantaneous linewidth narrowing, and the large-mode volume $Si_3N_4$ coil resonator provides a frequency reference for linewidth reduction and laser stabilization. **b** The frequency noise spectral energy (offset from the carrier) of the free-running ECTL ($S_f(\nu)$ blue) and

the noise reduction, particularly of the low-frequency noise components, resulting from stabilizing the ECTL to the 10 m coil reference cavity (orange). **c** Image of the hybrid-integrated ECTL, the RSOA is visible in the top left, the red-dashed lines highlight the $Si_3N_4$ waveguides, and gold pads indicate the metal heaters. **d** Schematic cross-section of the 80 nm thick $Si_3N_4$ low-loss waveguide platform. **e** Image of the 10 m coil resonator. "Photonic integrated beam delivery for a rubidium 3D magneto-optical trap" by Isichenko et al., used under CC BY 4.0/Panel **a** adapted from original[52].

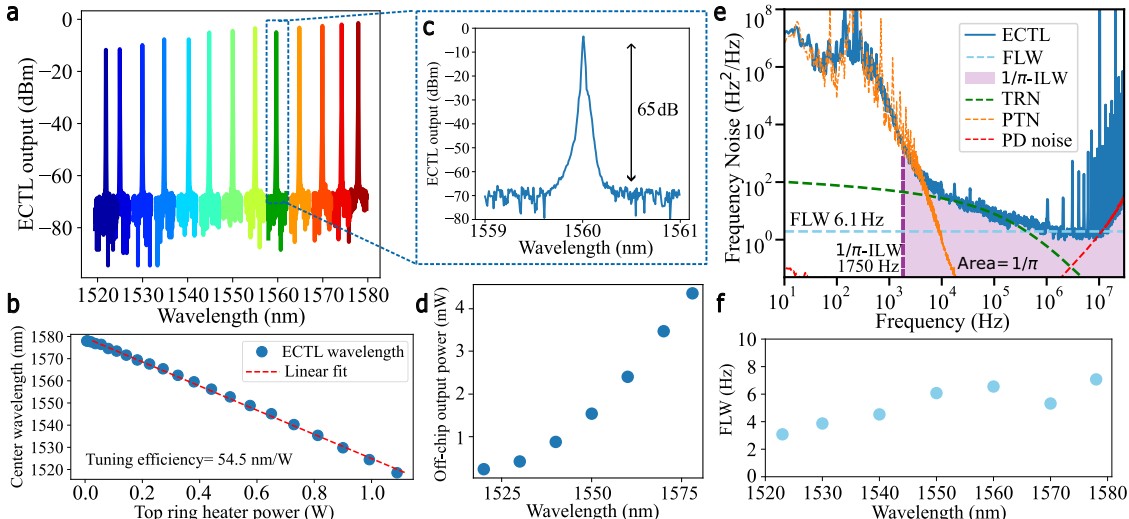

**Fig. 2 | External cavity tunable laser performance. a** Single mode laser output from 1520–1580 nm. **b** Thermo-optic tuning of the top ring of the external cavity tunable laser (ECTL). **c** ECTL operation at 1560 nm with a measured side mode suppression ratio (SMSR) of ~ 65 dB, measured on an optical spectrum analyser (OSA) with resolution bandwidth (RBW) <0.01 nm. **d** Fiber-coupled ECTL output power measured across the tuning range. **e** Frequency noise spectrum of the free-running ECTL at 1550 nm (blue, solid) was measured using a fiber-Mach Zehnder Interferometer (fiber-MZI) as an optical frequency discriminator (OFD). The blue dashed line plots the measured ECTL fundamental linewidth (FLW) of 6.1 Hz, and

the shaded purple region shows the area under the curve that contributes to the $1/\pi$ integral linewidth (ILW) of 1750 Hz. The green and orange dashed curves are simulated estimates of the thermorefractive noise (TRN) and photothermal noise (PTN) limits of the ECTL rings, and the red dashed curve is the OFD photodetector noise. The high frequency spurs at multiples of 1 MHz correspond to the free-spectral range of the OFD fiber-MZI and do not contribute to the integral linewidth calculation. **f** FLW of the ECTL measured across the 60 nm tuning range. Higher reflectivity of the Sagnac loop mirror and weaker ring-bus coupling may contribute to the decrease in FLW at shorter wavelengths.

to 2.6 µm wide to achieve ultra-low-propagation loss in the external cavity circuit (see "Methods"). The thermo-optically controlled ring resonators, with radii of 1998.36 and 2002.58 µm, have an FSR of ~126.5 pm and therefore a Vernier FSR of ~59.9 nm. The rings have an intrinsic-Q of 3.5 million (see Supplementary Note 2) and are designed to be over-coupled to reduce the lasing threshold, resulting in a loaded-Q of 0.65 million. The laser output waveguide is coupled to a lensed fiber.

The silicon nitride 10 m coil resonator (Fig. 1e) is a 25 mm × 25 mm chip consisting of a bus-coupled coil waveguide geometry (see Supplementary Note 3) with a measured propagation loss of 0.2 dB/m and an intrinsic-Q of ~200 million. The 10 m-long cavity reduces the TRN floor of the stabilized laser due to its large mode volume[42,48]. Additionally, the 10 m coil length provides a fine 20 MHz FSR with good fringe extinction ratio over more than 80 nm of bandwidth that provides an almost continuous range of lock frequencies across the widely tunable laser range of 60 nm. The frequency noise performance is measured using an unbalanced optical frequency discriminator (OFD) for offset frequencies >1 kHz and a ULE cavity-stabilized fiber frequency comb for offset frequencies of 1 kHz down to 1 Hz.

## ECTL performance

The ECTL performance is summarized in Fig. 2. We demonstrate a 60 nm-wide single-mode operating range, from 1518.5 to 1578 nm (Fig. 2a), with a SMSR of 65 dB (Fig. 2c) across the tuning range. The ECTL ring heater tuning efficiency is 54.5 nm/W per actuator (Fig. 2b). We measure a lasing threshold current at 1550 nm of 63 mA and a fiber-coupled output power ranging from 0.23 mW at 1520 nm to 4.37 mW at 1578 nm (Fig. 2d). The change in output power is primarily due to the wavelength dependence of the evanescent coupler in the Sagnac loop mirror. The frequency noise (FN) spectrum of the free-running, widely tunable ECTL without coil-resonator stabilization is shown for 1550 nm emission in Fig. 2e, indicating the fundamental and ILWs of the free-running laser at this wavelength. For intermediate offset frequencies, ~2–200 kHz, the FN closely follows the calculated TRN limit of the

ECTL rings (green-dashed curve). The laser noise reaches the white frequency noise (WFN) floor characteristic of the Lorentzian FLW at >1 MHz frequency offset. The periodic high-frequency spikes are from the fiber Mach-Zehnder interferometer (MZI) frequency noise measurement setup. Photothermal noise dominates at lower than 2 kHz and the MZI photodiode noise dominates at frequencies greater than 100 MHz. We measure a WFN floor at 1550 nm of 1.93 Hz²/Hz, corresponding to a 6.08 Hz (light-blue dashed line) FLW. The ILW of the free-running ECTL at this wavelength is calculated using $1/\pi$ the reverse integration method[40] to be 1.75 kHz and shown in the purple shaded region under the FN curve. Across the full tuning range, we measure the FLW of the free-running ECTL to be in the range of 3–7 Hz as summarized in Fig. 2f (see Supplementary Note 4 for the full dataset).

## Coil stabilized ECTL performance

We stabilize the ECTL to the 10 m long silicon nitride integrated coil reference cavity without an optical isolator between the laser and reference cavity using an EOM to generate locking sidebands of 10 – 20 MHz and a PDH error signal fed back directly to the ECTL gain chip current using a Vescent D2-125 laser servo. The ECTL takes on the FN characteristics of the coil resonator for frequency offsets up to the PDH locking bandwidth. Notably, the high internal quality factor of the ECTL, due to the low-loss ring resonators, provides high resilience to optical feedback and removes the need for an optical isolator between the ECTL and coil reference cavity to operate the stabilization lock. Eliminating the need for an isolator between the two components is important for future integration of the laser and reference cavity onto a single silicon nitride chip.

The stabilized-ECTL frequency noise, linewidth, and ADEV are measured using two independent techniques. For FN above 3 kHz frequency offset, we use an asymmetric fiber MZI OFD. Below 3 kHz offset down to 1 Hz we measure the noise of a heterodyne beatnote of the ECTL with an optical frequency comb that has been locked to a stable reference laser (SRL). The beatnote is then measured on a precision frequency counter (See "Methods" for more details). The FN of

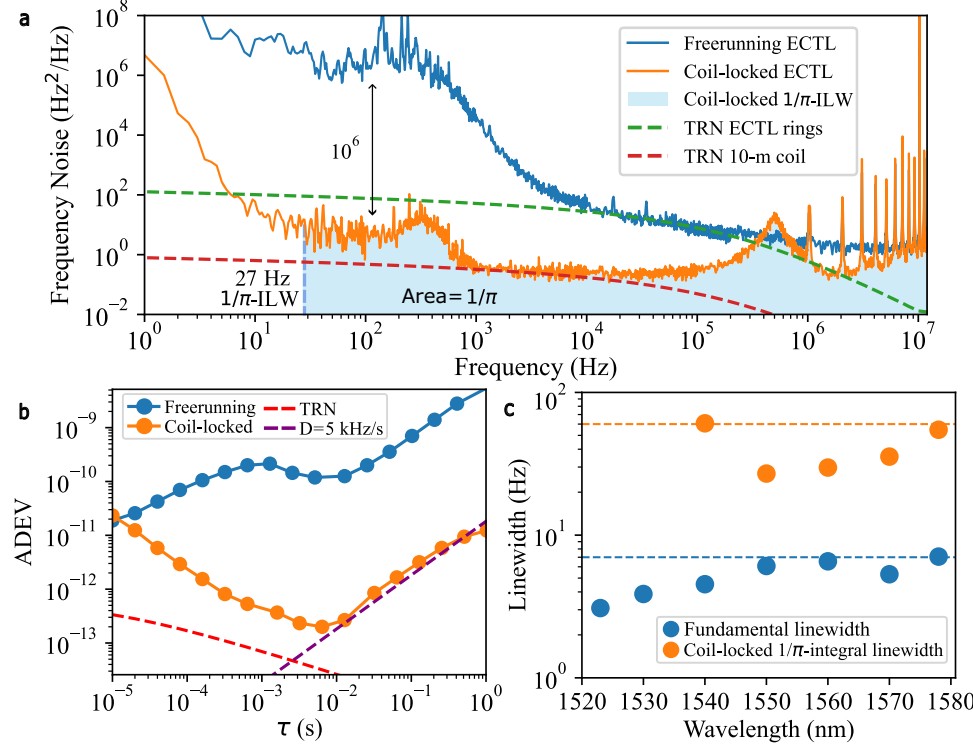

**Fig. 3 | Coil-stabilized ECTL frequency noise, stability, and fundamental and integral linewidths across tuning range. a** An example frequency noise (FN) spectrum of the free-running (blue) and coil-locked (orange) external cavity tunable laser (ECTL) at 1550 nm shows >6 orders of magnitude reduction in FN at low frequency offsets. The shaded light-blue area corresponds to the $1/\pi$ reverse integral linewidth (ILW) of the locked laser measured at 27 Hz. The minimum FN of the locked laser is measured at 0.12 Hz²/Hz at 16 kHz frequency offset. The Pound–Drever–Hall (PDH) lock servo bump is indicated at 0.5 MHz. **b** Allan deviation (ADEV) of the free-running (blue) and coil-locked (orange) ECTL demonstrating $1.8 \times 10^{-13}$ at 6.4 ms and 5 kHz/s drift. **c** The measured fundamental linewidth (FLW) (blue) across the 60 nm tuning range and coil-locked ILW (orange) across a 40 nm tuning range. The ILW measurement was limited by the beatnote measurement below 1540 nm. In addition, the coil-lock at 1520 nm and below was limited due to low extinction ratio (ER). The dashed lines indicate the highest measured value of 7 Hz for FLW (blue) and 60 Hz for ILW (orange).

the coil-locked ECTL operating at 1550 nm is plotted in Fig. 3a. The stabilized ECTL (orange) has a FN reduction of more than 6-orders of magnitude at low frequency offsets compared with the free-running laser (blue) and reaches the TRN limit of the 10 m coil resonator (red, dashed) between 1 and 100 kHz offset, before sloping upwards due to the servo bump corresponding to the PDH locking bandwidth at around 0.5 MHz. The measured $1/\pi$ ILW of the stabilized-ECTL at 1550 nm is 27 Hz, reduced from 1750 Hz for the free-running laser, an ILW reduction of 65×. The minimum FN is 0.12 Hz²/Hz at 16 kHz, and the corresponding fractional frequency stability, or ADEV (Fig. 3b), has a minimum of $1.8 \times 10^{-13}$ at 6.4 ms and a drift of 5 kHz/s. The fundamental and ILWs of the stabilized-ECTL at various points across the tuning range are plotted in Fig. 3c, see Supplementary Note 4 for more details.

### ECTL resilience to optical feedback

The ECTL resilience to optical feedback is demonstrated by operating the PDH lock to the 10 m coil reference cavity without an isolator— marking the first such demonstration for a Vernier-style laser and a key step toward fully integrated stabilized lasers on-chip. To further investigate this under controlled conditions, we utilized an optical feedback measurement setup[43–45] (see Fig. 4a). Using a fiber circulator and a variable optical attenuator (VOA) we precisely control the feedback level back to the laser while accounting for circuit losses to determine the power reaching the laser. A small fraction of the ECTL output is diverted to measure the power and the frequency noise of the laser, the results of which are plotted in Fig. 4b. Compared to a commercial III–V DFB laser that experiences FN degradation and

coherence collapse at a feedback level of −40 dB[43–45], the ECTL maintains single-mode operation across all tested feedback levels, up to −10 dB, with no degradation in the frequency noise. This represents a 30 dB improvement in resilience to optical feedback over a commercial DFB laser and enables the isolator-free coil resonator stabilization.

We observe in Fig. 4b that the measured frequency noise of the ECTL decreases with increasing optical feedback. This is likely due to the long coherence time and strong mode selectivity of the ECTL where photons that re-enter from the external feedback loop remain phase-coherent with the intracavity field, effectively increasing the intracavity photon number and reducing quantum noise. Additionally, the extended fiber feedback circuit increases the overall optical mode volume, which suppresses TRN, leading to a lower frequency noise floor.

### Discussion

We demonstrate a precision ultra-low instantaneous and ILW widely tunable stabilized hybrid integrated ECTL locked to an integrated coil reference cavity that achieves record frequency and phase noise for hybrid-integrated widely tunable lasers. The coil-locked ECTL realizes 6 Hz FLW with 27 Hz ILW and $1.8 \times 10^{-13}$ fractional frequency stability at 1550 nm, and 3–7 Hz fundamental with sub-60 Hz ILWs measured across the tuning range. These linewidths are the lowest to date over the widest tuning range for integrated chips. Furthermore, we demonstrate isolation-free stabilization of the ECTL to the coil cavity, enabled by the high Q of the ECTL cavity. The 10 m coil has an FSR of 20 MHz, which allows locking and stabilization at almost all wavelengths across the tuning range, a distinct advantage over low FSR bulk optic reference cavities.

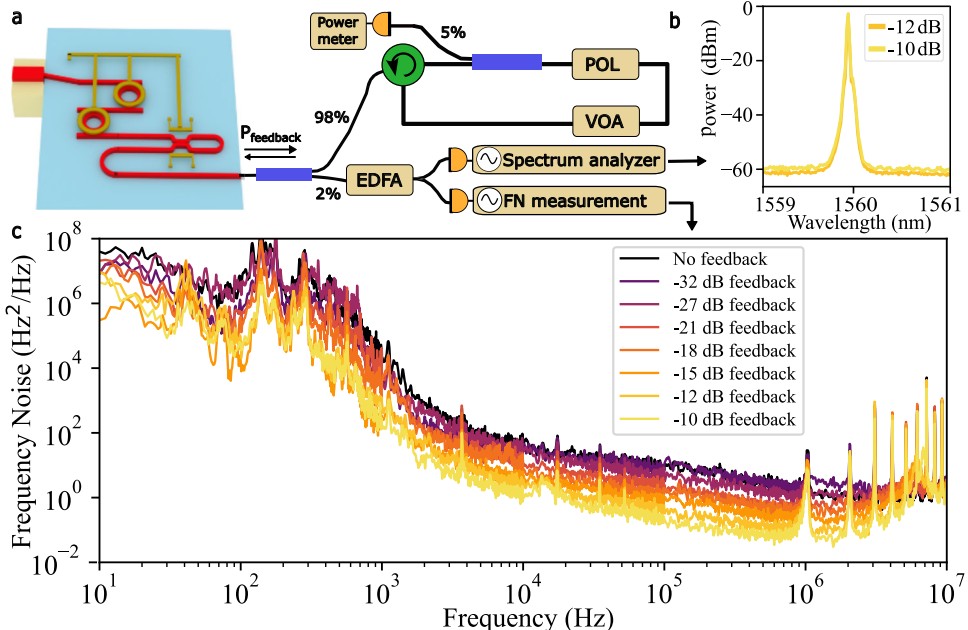

**Fig. 4 | ECTL feedback measurements. a** Schematic of the experimental setup for measuring the effect of optical feedback on the external cavity tunable laser (ECTL). The laser output power is initially split in two: the majority for feeding back to the laser and a small portion to measure the laser frequency noise (FN). A variable optical attenuator (VOA) sets the feedback power level that is monitored with a power meter. Polarization (POL) paddles ensure the returned optical field is aligned properly with the silicon nitride waveguide. The FN is measured with an optical frequency discriminator (OFD) and an optical spectrum analyzer (OSA) monitors

for the onset of coherence collapse. **b** OSA trace of the ECTL output under the two highest feedback levels of −12 and −10 dB. **c** Frequency noise plots of the free-running ECTL operating at 1560 nm under different optical feedback levels ranging from −32 to −10 dB. The maximum feedback level is limited by the fiber-chip insertion loss and all fiber splitter and connector losses. The inset plots the OSA traces for ECTL output under the two highest optical feedback levels showing single-mode operation.

**Table 1 | Comparison of low linewidth hybrid-integrated tunable lasers**

| Platform | Laser type | λ (nm) | FLW (Hz) | ILW (Hz) 1/π | ADEV (@ 1 ms) | Tuning (nm) | Output power (mW) | SMSR (dB) | Optical isolation (dB) |
|---|---|---|---|---|---|---|---|---|---|
| Si₃N₄[16] | SIL | 1550 | 0.04 | 236[b] | ... | 0.8 | 0.3 | 60 | ... |
| Si₃N₄[58] | SIL | 1550 | 3.8 | 4715[b] | ... | ... | 10.5 | 65 | ... |
| Si₃N₄[59] | SIL | 1550 | 3 | 1560[b] | ... | ... | ... | 54 | ... |
| Si₃N₄[18] | SIL | 780 | 0.74 | 864 | ... | 2 | 2 | 36 | ... |
| Si₃N₄[60] | SIL | 785 | 700 | 50,173[b] | ... | 12 | 10 | 37 | ... |
| Si₃N₄[23] | EDBR | 1550 | 320 | 47,466[b] | ... | ... | 24 | 55 | ... |
| Si[61] | ECTL: 3 ring | 1550 | 220 | 33,246[b] | ... | 110 | 3 | 50 | ... |
| Si[28] | ECTL: 3 ring | 1550 | 95 | 9237[b] | ... | 120 | 1.5 | 60 | ... |
| Si₃N₄[39] | ECTL: 3 ring | 1550 | 40 | 87,844[c] | ... | 70 | 23 | 60 | ... |
| Si₃N₄[62] | ECTL: 2 ring | 1550 | 2200 | 57,526[b] | ... | 120 | 24 | 63 | ... |
| Si₃N₄[36] | ECTL: 2 ring | 1550 | 750–4000 | 31,614[b] | ... | 172 | 26 | 68 | ... |
| Si₃N₄[63] | ECTL: 2 ring | 852 | 65 | 6770[b] | ... | 15 | 25 | 50 | ... |
| Si₃N₄[26] | ECTL: 2 ring | 1550 | 6–9.8[d] | 2350 | ... | 40 | 4.8 | 64 | ... |
| **Si₃N₄**[a] | **ECTL (2 ring) + coil** | **1550** | **3–7**[d] | **27** | **1.8E-13@6.4 ms** | **60** | **4.4** | **65** | **−30**[e] |

[a]Bold text signifies results from this work.
[b]Not reported in manuscript: calculated from published FN data.
[c]ILW limited by available FN data.
[d]Measured across tuning range.
[e]Relative to a typical commercial III–V DFB laser.

We compare our results to the state of the art in Table 1 and Fig. 5 with other hybrid-integrated low noise lasers. The table is sparsely populated in several columns since this work represents one of the few that report the 1/π ILW, which is important for many precision applications. When not available directly in the publication, we have calculated the ILW from the available FN data in that publication. We

believe this comparison highlights the unique properties of the hybrid-integrated ECTL locked to the integrated coil reference cavity. For example, we report both the fundamental and integral linewidths over a full tuning range. The FLW is 1.5× lower and the 1/π integral linewidth 3-orders of magnitude lower with a tuning range 1.5× larger than state of the art integrated ECTLs[26]. Additionally, this work is the first to

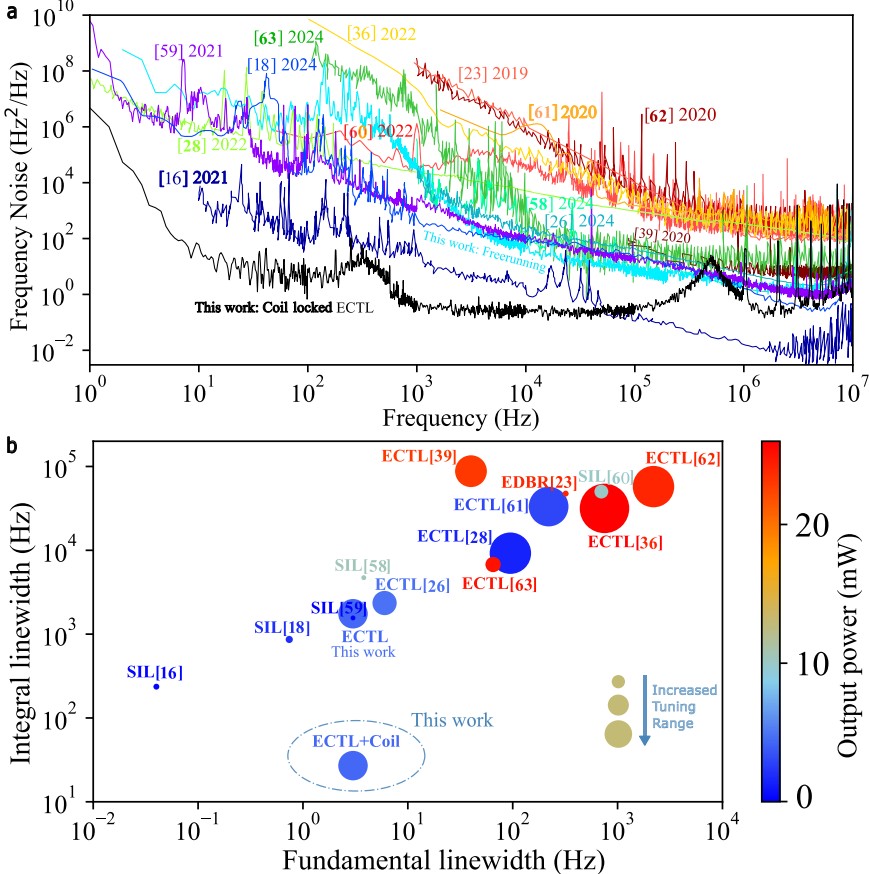

**Fig. 5 | Comparison of low linewidth hybrid-integrated lasers summarized in Table 1. a** Frequency noise plots comparing hybrid-integrated low noise lasers. **b** Fundamental vs. $1/\pi$ integral linewidths. The bubble size represents the tuning range, and the color heat map represents output power.

report an ADEV, 1.6E-13 at 6.4 ms, and isolator-free operation in a Vernier-style hybrid laser.

Since both the ECTL and coil resonator are fabricated in the same 80 nm thick, CMOS-compatible $Si_3N_4$ platform, these results present a clear path forward towards realizing a fully integrated, chip-scale stabilized laser that combines narrow instantaneous linewidth with low frequency drift in one device. In future versions, the EOM used in this experiment can be eliminated by adding a PZT-on-$Si_3N_4$ integrated double sideband modulator that can achieve locking bandwidths up to 20 MHz[49], or by employing modulation-free stabilization techniques[49,50]. Thermal and PZT actuators have been demonstrated in this $Si_3N_4$ platform without affecting waveguide loss, independent of wavelength, and operate from DC out to 10 s of MHz[35] and could also be used to directly tune the intracavity ECTL rings rather than feeding the PDH error signal back to the gain chip current. Additionally, the core components of this stabilized laser design have been demonstrated across the visible to SWIR range[51] making this hybrid-integrated stabilized laser design a potential laser source for silicon nitride photonics that support atomic and quantum applications[52–54].

Further experiments and design improvements will yield better laser performance in terms of tunability, variation in integral and FLW, and higher output power. For example, full control of the Sagnac loop mirror would provide an adjustment of the laser output power, and therefore also the intracavity power, allowing the user to optimize for higher output power versus lower FLW. Additionally, loaded-Qs as high as 100 million have been demonstrated in this platform, and since the modified Schawlow–Townes laser linewidth will decrease as $1/Q^2$, we predict that future devices could achieve even lower FLWs. Future designs can also include coil resonators longer than 10 m to increase the mode volume and further reduce the

TRN and ILW, as well as incorporate tunable coupling[55] to optimize the laser locking conditions across a wide range of wavelengths using a single resonator. For applications that require high optical output power, additional gain blocks can be added and operated in parallel with a shared high-Q silicon nitride external cavity[25], where output powers >100 mW have already been demonstrated in a dual-gain hybrid-integrated laser[46]. Other pathways to increasing the output power are to incorporate on-chip amplifiers[56,57] or through injection-locked amplification.

## Methods
### Fabrication process
The lower cladding consists of a 15 μm-thick thermal oxide grown on a 100 mm diameter, 1 mm-thick silicon wafer substrate. The main waveguide layer is an 80 nm thick stoichiometric $Si_3N_4$ film deposited on the lower cladding using low-pressure chemical vapor deposition. A PAS 5500 ASML deep ultraviolet (DUV) stepper was used to pattern a DUV photoresist layer. The high-aspect-ratio waveguide core is formed by anisotropically dry etching the $Si_3N_4$ film in a Panasonic E626I Inductively Coupled Plasma-Reactive Ion etcher using a $CHF_3/CF_4/O_2$ chemistry. After the etch, the wafer is cleaned using a standard RCA cleaning process. A 5 μm-thick silicon dioxide upper cladding layer was deposited using plasma-enhanced chemical vapor deposition with tetraethoxysilane as a precursor. This is followed by a final two-step anneal at 1050 °C for 7 h and 1150 °C for 2 h which is an optimized anneal process for our waveguides.

### RSOA coupling
The gain chip is coated with 90% reflectivity on the side opposite the silicon nitride PIC and an antireflection material on the near side with a

reflectivity of 0.005%. It is then mounted on a temperature-controlled copper block for heat-sinking. The gain chip is wire bonded to a PCB that screws onto the copper block for external electrical control of the gain chip. The SOA has an angled facet of 5.6° that requires, based on Snell's Law, the $Si_3N_4$ waveguide to be angled by 13.1° to best match the beam propagation direction. The $Si_3N_4$ waveguide is initially 18 μm-wide to achieve an estimated optimal mode overlap of 52%. We estimate that the actual coupling loss is around 4−5 dB. The $Si_3N_4$ PIC sits atop its own temperature-controlled mount and the RSOA is edge-coupled to the ECTL input waveguide.

### Frequency noise measurements

The coil-stabilized ECTL frequency noise, linewidth, and stability is measured using two independent techniques[18,40]. For FN above 3 kHz frequency offset, we use an unbalanced fiber-MZI with a 1.026 MHz FSR as an OFD and measure the self-delayed homodyne FN signal on a high-speed balanced photodetector. Below 3 kHz offset, the OFD fiber-MZI noise can become dominant, so we instead photomix the ECTL with a SRL and measure the heterodyne beatnote signal with a Keysight 53230 ADEV precision frequency counter. The SRL system consists of a single-frequency Rock fiber laser PDH-locked to a Stable Laser Systems (SLS) 1550 nm ULE cavity that delivers Hz-level linewidth and ~0.1 Hz/s frequency drift. Additionally, we use a Vescent self-referenced fiber frequency comb with a 100 MHz $f_{rep}$ and lock it to the SLS to extend the stability of the SRL system to many wavelengths and enable accurate close-to-carrier FN measurements across the ECTL spectrum.

### Optical feedback measurements

The optical feedback measurement, illustrated in Fig. 4a, utilizes a fiber circulator and a VOA to control the optical feedback level. The power returning to the PIC is polarization sensitive and must be adjusted to maximize the optical feedback before making any measurements. The power returned to the ECTL is monitored by tapping 5% of the feedback power and measuring on a power meter. Another small fraction of the laser power is diverted to an EDFA and then to an OFD setup to measure the frequency noise of the laser under different feedback conditions. The full fiber feedback circuit is ~25 m long. All losses in the feedback circuit must be accounted for to most accurately determine the actual feedback power to the laser. The highest level of optical feedback in our experiment was −10 dB, limited primarily by the optical losses due to fiber-to-chip coupling and the connectors and splitters in the feedback loop.

## Data availability

The data that support the plots within this paper and other findings of this study are available from the corresponding author on request.

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

## Acknowledgments

We acknowledge Henry Timmers at Vescent for help with the setup of the fiber frequency comb and Mark W. Harrington at UCSB for help with the SRL system and Karl Nelson of Honeywell for help fabricating the coil resonator. This work is based in part by funding from DARPA GRYPHON award HR0011-22-2-0008. The views and conclusions contained in this document are those of the author(s) and should not be interpreted as representing the official policies of DARPA or the U.S. government.

## Author contributions

D.A.S.H., D.J.B., K.L., D.B. and A.I. prepared the manuscript. D.B. designed and fabricated the ECTL PICs. D.A.S.H. and A.I. built the ECTL. D.A.S.H. characterized the ECTL performance. K.L. designed and tested the coil resonator. D.A.S.H. and K.L. performed all laser locking and frequency noise experiments. A.I. built the SRL and optical frequency comb frequency noise measurement system. D.J.B. supervised the project. All authors participated in the writing of the manuscript.

## Competing interests

D.J.B.'s work has been funded by ColdQuanta d.b.a. Infleqtion. D.J.B. has consulted for Infleqtion and received compensation, is a member of the scientific advisory council, and owns stock in the company. D.A.S.H., D.B., K.L., and A.I. declare no competing interests.
