## [Transparent Peer Review file · Nature Communications]

Hybrid integrated ultra-low linewidth coil stabilized isolator-free widely tunable external cavity laser

Corresponding Author: Professor Daniel Blumenthal

Version 1:

Reviewer comments:

Reviewer #1

(Remarks to the Author)

D.A.S. Heim et al. present work on a “Hybrid integrated ultra-low linewidth coil stabilized isolator-free widely tunable external cavity laser” consisting of a chip based external cavity tunable laser that is locked to a 10m long chip based spiral resonator on a separate Si_3N_4 photonic chip. The work is quite similar to earlier work by the same authors [Ref. 30], replacing the commercial low noise laser with a tunable Vernier filter-based laser and increasing the length of the coil. In this way the ultra-low noise laser system inherits the long tuning range functionality from the Vernier laser. In addition the authors achieve operation of the system without an optical isolator. Operation of ultra-low noise lasers without isolators has recently garnered much attention in the framework of self-injection locked lasers [1,2] but was not investigated for Vernier-style low noise lasers yet. Whether the mere combination of these advancements and/or the admittedly very strong performance of the presented laser system merits publication in Nature Communications is left to the editors.

Fig1 shows the authors vision of a fully integrated version of their laser with the Vernier laser, the EOM and the coil waveguide resonator, but not the photonic chip based laser that is discussed in the remainder of the manuscript. The misleading Figure 1 hence should be replaced / combined with Figure 3, which shows the actual system and measurement setup. It is also not clear why the measurement setup Fig3 is placed after the measurement results Fig2 except for the observation that the middle part of Fig1 and the left part of Fig3 are nearly identical. In addition, the schematic illustration on top of Fig 1 is unclear.

Another point is that the authors’ claim in the abstract of “leveraging 45 dB internal isolation” is dubious, since any backreflection of laser light would be on resonance with the Vernier filter and enter the laser. Indeed, a suppression of the feedback response of 45 dB is not demonstrated in the manuscript, only estimated from the optical feedback parameter. The measurements in Fig. 5 suggest that self-injection locking (SIL) between the laser and the external fiber loop feedback circuit takes place. As SIL implies a frequency pulling effect on the laser, such optical feedback may hence affect the frequency stability of the laser even well below the level required to induce laser instability and chaos.

Do the authors observe any frequency pulling, noise reduction or self-injection locking of the laser to the ring cavity when the two chips are connected? Injection locking of a kHz laser to an ultra-low noise cavity was demonstrated in Ref 44 and may also work in the case here.

What is the modulation frequency of the EOM for PHD locking?

[1] Xiang, C., Jin, W., Terra, O. et al. 3D integration enables ultralow-noise isolator-free lasers in silicon photonics. Nature 620, 78–85 (2023). <https://doi.org/10.1038/s41586-023-06251-w>

[2] White, A.D., Ahn, G.H., Luhtaru, R. et al. Unified laser stabilization and isolation on a silicon chip. Nat. Photon. 18, 1305–1311 (2024). <https://doi.org/10.1038/s41566-024-01539-3>

Reviewer #2

(Remarks to the Author)

The paper demonstrates a hybrid integrated coil-stabilized laser with record-low integral linewidth (27-60 Hz) across a 60-nm tuning range. It utilizes a Vernier-based external cavity for laser linewidth reduction and tunability enhancement, and an ultra-high-Q coil cavity for frequency stabilization, both fabricated in the same Si₃N₄ integrated platform.

Si₃N₄-based external cavity lasers have been developed for many years, and it is well known that high-Q microrings can be used as reference cavities in the PDH technique. The authors have pushed the state of the art by combining the above two techniques, enabling the external cavity laser to be locked to the coil cavity across various resonance within the full tuning range. The proposed laser is promising for many applications such as ultra-precise metrology, microwave photonics, and quantum optics. Given the record-breaking performance, solid experimental results and future potential impact, I recommend this paper to be accepted by Nature Communications after addressing the following issue.

1. My main concern is about the design and performance of external cavity laser. I understand that the authors employ large rings for sufficient linewidth reduction, but it may be difficult to support single-mode lasing. Based on the parameters given in the manuscript, I simulated the reflection spectrum of the Vernier-based external cavity by simply multiplying the transmissions of the two rings. The simulated adjacent sidelobe suppression of the external cavity is less than 0.5 dB. I am concerned that such a low level of sidelobe suppression may not be sufficient to ensure stable single-mode lasing, making mode hopping likely to occur. I hope the authors provide simulation results of the Vernier-based external cavity and address this issue.

2. The output power below 1 mW is significantly lower than over-10-mW power in other external cavity lasers. Please give an explanation and potential solutions, if any.

3. As shown in Fig. 2(d), lasing at a shorter wavelength requires a high heating power on the ring, subsequently leading to a higher thermorefractive noise. However, in Fig. 2(d), the fundamental linewidth is decreased with the shorter wavelength. Please explain this issue.

4. In laser noise measurement, the authors give the estimated TRN and PTN curve in Supplementary Fig. 3, which fit the measured laser frequency noise curve well. Could the authors give more details about the calculation method and the corresponding estimation parameters, especially the PTN calculation that is rarely mentioned in the previous papers. Moreover, I suggest them to add this PTN curve in Fig. 2(e) in the manuscript.

5. In the laser stabilization demonstration, the PDH error signal is directly fed into the RSOA. On account of the high-Q property of the external cavity, could this minor phase shift lead to large laser power fluctuation? In addition, does the long photo lifetime of the external cavity laser limit the PDH locking bandwidth? Please comment on these issues, and a minor personal suggestion is to add the PDH residual noise curve in Fig. 4 (a).

6. The laser feedback measurement results are strange. Commonly the laser performance deteriorates with the increased feedback light power, but the experiment shows the opposite result. As the authors claimed, it is possible that the laser enters a self-injection locking regime under specific phase-matching conditions. However, I think the experiment results do not validate the feedback insensitivity of the laser. Such a phenomenon should be avoided in the experiment, because the feedback light with random phase disrupts the oscillation in the practical scenarios. Moreover, Sidelobe suppression ratio is a critical indicator in feedback measurement that should be characterized. I suggest the authors to repeat this experiment and refer to the methodologies outlined in two papers listed below.

[1] Xiang, Chao, et al. "3D integration enables ultralow-noise isolator-free lasers in silicon photonics." *Nature* 620.7972 (2023): 78-85.

[2] Tang, Liwei, et al. "A method for improving reflection tolerance of laser source in hybrid photonic packaged micro-system." *IEEE Photonics Technology Letters* 33.9 (2021): 465-468.

7. The authors have done a good job in comprehensively comparing their work with the previous results, and Fig. 6 is very clear and valuable. Nevertheless, it is inappropriate to limit the comparison to hybrid integrated lasers while excluding some high-performance heterogeneous integrated lasers (references are listed below). I recommend that the authors expand their comparison to include chip-scale lasers, particularly by adding Ref. [3], which demonstrates a state-of-the-art integral linewidth of 1 Hz.

[3] Guo, Joel, et al. "Chip-based laser with 1-hertz integrated linewidth." *Science advances* 8.43 (2022): eabp9006.

[4] Morton, Paul A., et al. "Integrated coherent tunable laser (ICTL) with ultra-wideband wavelength tuning and sub-100 Hz Lorentzian linewidth." *Journal of Lightwave Technology* 40.6 (2022): 1802-1809.

Reviewer #3

(Remarks to the Author)

The manuscript presents a hybrid-integrated external cavity tunable laser (ECTL) stabilized to an integrated silicon nitride coil resonator. The authors claim record-low fundamental linewidths and improved noise performance without the need for an optical isolator. The paper highlights the potential for compact and robust photonic integration in ultra-narrow linewidth lasers for precision applications. My concerns are the following:

- The work primarily builds on previously published results on external cavity lasers and hybrid integration techniques. While the authors claim significant improvements in linewidth and stability, the advances are incremental rather than groundbreaking.

- The isolation effect is primarily attributed to the high-Q nature of the optical rings and the low-loss SiN platform by the

same group has already been reported. Repurposing the high-Q ring resonators for a different application does not constitute a novel innovation and does not justify the claim of a significant advancement.

- What is the reason for quite high-frequency noise at lower frequency offsets?
- The EOM and spiral cavity are not integrated, which could have added to the novelty of the manuscript.
- The impact of the paper is very similar to Guo et al., Science Advances (2022) (<https://www.science.org/doi/10.1126/sciadv.abp9006>) with little differentiation.
- The manuscript claims that the ECTL inherently provides 45 dB of isolation. However, this assertion is not well supported by experimental comparisons with commercially available isolators or experimental data.

Given the incremental nature of the work and lack of novelty, I do not recommend this manuscript for publication in Nature Communications.

Version 2:

Reviewer comments:

Reviewer #1

(Remarks to the Author)

D.A.S. Heim et al. have cured my two concerns with the manuscript relating to figures 1 and 3 and the claim of 45 dB isolation claim. From a technical point of view, the manuscript could be published. Whether the novelty of the work merits publication in Nature Communications is left to the editors.

Reviewer #2

(Remarks to the Author)

I think the revised manuscript has adequately addressed all concerns raised during the review process and is now acceptable for publication.

Reviewer #3

(Remarks to the Author)

I appreciate the authors' detailed rebuttal and clarifications regarding the technical contributions and distinctions from prior work. There is no doubt that the manuscript reports a high-performance hybrid integrated ECTL with impressive metrics, particularly in achieving a record-low integral linewidth and in demonstrating potential pathways toward a fully stabilized on-chip laser.

However, I maintain that while the laser's performance represents an incremental advancement, the core novelty of a fully integrated on-chip stabilized laser remains a future goal rather than a realized achievement in this work. As the authors themselves acknowledge, the spiral resonator is not integrated on the same chip as the ECTL, and the system still relies on discrete components such as the external electro-optic modulator (EOM). While the authors suggest feasible paths for future integration, such as using PZT-on-Si₃N₄ or modulation-free stabilization techniques, these remain prospective developments.

The comparison with Guo et al. is appreciated, and the authors correctly point out important distinctions in tuning range, reference cavity integration, and use of isolators. Nevertheless, it should be noted that their own spiral resonator is not co-integrated on the same photonic chip as the laser, and therefore, the claim of achieving a fully on-chip stabilized laser, which is central to the manuscript's novelty, is not yet substantiated in this implementation.

Additionally, the wide tunability achieved via Vernier tuning, although valuable, is not novel in itself, as this technique has been well demonstrated in previous literature across various platforms.

I believe the manuscript makes a valuable technical contribution and would be of interest to the photonics and laser stabilization community. However, given that the main novelty remains partially realized, I feel that the manuscript would be more appropriately positioned in a specialist journal with a focus on integrated photonics or laser engineering rather than in Nature Communications, which typically prioritizes transformative advancements with immediate and broad impact.

Appeal Response to Nature Communications

“Ultra-low linewidth coil stabilized widely tunable isolator-free hybrid integrated external cavity laser”

Manuscript NCOMMS-24-78214

We would like to thank the reviewers for their valuable time, questions, feedback, and comments. The points raised by the reviewers and their active involvement have resulted in a greatly improved manuscript as well as more effective communication of the significance of these results to a broad audience.

We have provided below a detailed response on a point-by-point basis for each reviewer’s comment (*blue*) with our response (*black*) and an excerpt of the updated text (*red*).

Reviewer #1 (Remarks to the Author):

D.A.S. Heim et al. present work on a “Hybrid integrated ultra-low linewidth coil stabilized isolator-free widely tunable external cavity laser” consisting of a chip based external cavity tunable laser that is locked to a 10m long chip based spiral resonator on a separate Si₃N₄ photonic chip.

The work is quite similar to earlier work by the same authors [Ref. 30], replacing the commercial low noise laser with a tunable Vernier filter-based laser and increasing the length of the coil. In this way the ultra-low noise laser system inherits the long tuning range functionality from the Vernier laser.

We agree with the Reviewer that in this work we demonstrate, for the first time, an ultra-low noise laser system that includes the long tuning range functionality of a Vernier laser. However, we respectfully disagree that this work is similar to our previous work. We believe this is a significant result and different from what has been reported before. In the previous work we did not report our own silicon nitride integrated widely tunable external cavity laser, with Hz level fundamental LW, a 60 nm tuning range, and in the same waveguide design and process as a 10 meter coil resonator. In the new work, it is not a matter of only lengthening the coil and achieving record high Q for a 10 meter coil - the long 10 meter coil has fringes every 20 MHz allowing the tunable laser to be locked at many wavelengths across a broad 60 nm tuning range. This is not possible with short coils (that have large FSR) or microrods (which also have large FSR) or using SIL lasers that are not broadly tunable. This is a significant differentiator and advance in the field. Nor did we report in the prior work both the fundamental and integral linewidths, nor did we report before the performance across a wide tuning range, nor did we report on the reflection tolerance and show that the laser could be coupled to the coil resonator without an optical isolator. This work is new and significant. The fact that this result has the laser and coil resonator in the same platform and Ref [30] has a commercial laser and shorter coil, and does not reach any of the performance aspects described in this work. This work presents widely tunable, record low fundamental and integrated linewidths, for the laser and coil reference, as a significant step towards a fully integrated stabilized laser in a way that the Ref [30] does not.

[30] K. Liu, N. Chauhan, J. Wang, A. Isichenko, G. M. Brodnik, P. A. Morton, R. O. Behunin, S. B. Papp, and D. J. Blumenthal, "36 Hz integral linewidth laser based on a photonic integrated 4.0 m coil resonator," *Optica* **9**(7), 770 (2022).

In addition the authors achieve operation of the system without an optical isolator. Operation of ultra-low noise lasers without isolators has recently garnered much attention in the framework of self-injection locked lasers [1,2] but was not investigated for Vernier-style low noise lasers yet.

[1] Xiang, C., Jin, W., Terra, O. et al. 3D integration enables ultralow-noise isolator-free lasers in silicon photonics. *Nature* 620, 78–85 (2023). <https://doi.org/10.1038/s41586-023-06251-w>

[2] White, A.D., Ahn, G.H., Luhtaru, R. et al. Unified laser stabilization and isolation on a silicon chip. *Nat. Photon.* 18, 1305–1311 (2024). <https://doi.org/10.1038/s41566-024-01539-3>

We agree and emphasize the Reviewer’s comment that this is the first demonstration of isolator-free operation of an integrated ultra-low noise Vernier-style laser. We have added refs. [1] and [2] to the manuscript for the aspect related to resilience to optical feedback with high Q.

We want to emphasize that our work is very different than Refs [1-2]. Most importantly, neither [1] nor [2] addresses the integral linewidth of the laser nor do they demonstrate such over a wide tuning range. Our result is the first to address achieving both low fundamental and low integral linewidths in an integrated device: a very important distinction for many applications. Also, neither [1] nor [2] are widely tunable (they are both < 1 nm), whereas our laser tunes across 60 nm. Also important for many applications. This combination of wide tunability and ultralow fundamental and integral linewidth have not been demonstrated before. Furthermore, Ref [2] relies on a nonlinear frequency shift to demonstrate resilience to optical feedback. This nonlinear shift requires high optical intensities and is an intensity dependent isolation which is why their demonstration utilizes a pulsed signal rather than a CW experiment like we and Ref [1] have demonstrated. This is not a robust approach to generate the isolation needed to build out chip-scale laser systems and will not scale well for real world applications.

Whether the mere combination of these advancements and/or the admittedly very strong performance of the presented laser system merits publication in *Nature Communications* is left to the editors.

We thank the Reviewer for their recognition of the “strong performance of the presented laser system.” We believe that this performance in a Vernier-style laser combined with the isolator-free operation and path towards a full stabilized laser on chip makes the result worthy of publication.

Fig1 shows the authors vision of a fully integrated version of their laser with the Vernier laser, the EOM and the coil waveguide resonator, but not the photonic chip based laser that is discussed in the remainder of the manuscript. The misleading Figure 1 hence should be replaced / combined with Figure 3, which shows the actual system and measurement setup.

We thank the Reviewer for this comment. The intent was not to mislead the reader but to present a vision figure of an ultra-low noise tunable stabilized laser on a single chip to help motivate the experiments demonstrated in the manuscript. We clearly state that Fig. 1 is a vision figure, and such vision figures are a regularly used to introduce experimental work. We refer to two such examples published in *Nature* [1] and *Nature Photonics* [2], respectively, where the first figure includes an illustrative example of a chip not actually fabricated as part of the work discussed in the manuscript (see figure below). Please see below on the left and right from each of these papers, showing the vision at the beginning of the manuscript, which was not actually demonstrated in the *Nature* and *Nature Photonics* papers.

However, we take the reviewers comments seriously and have removed the vision figure from the main manuscript and have moved the experimental setup from Fig. 3 into Fig. 1. The new Fig. 1 only includes a schematic of the experimental demonstration and images of the chips used in the results as well as a more clear description of the noise reduction properties the experimental work.

Examples of vision figures from two Nature publications that were not actually demonstrated in the papers: [1] Xiang, C., et al. 3D integration enables ultralow-noise isolator-free lasers in silicon photonics. Nature 620, 78–85 (2023). [2] Corato-Zanarella, et al. “Widely tunable and narrow-linewidth chip-scale lasers from near-ultraviolet to near-infrared wavelengths”, Nature Photonics (2023)

Update Figure 1.

It is also not clear why the measurement setup Fig3 is placed after the measurement results Fig2 except for the observation that the middle part of Fig1 and the left part of Fig3 are nearly identical.

We have moved the experimental setup into Fig. 1 and removed Fig. 3. The schematic of the frequency noise experimental setup has been moved to the Supplementary Fig. 3a.

In addition, the schematic illustration on top of Fig 1 is unclear.

We have removed the schematic illustration on top of Fig. 1.

Another point is that the authors’ claim in the abstract of “leveraging 45 dB internal isolation” is dubious, since any back reflection of laser light would be on resonance with the Vernier filter and enter the laser.

Indeed, a suppression of the feedback response of 45 dB is not demonstrated in the manuscript, only estimated from the optical feedback parameter.

We agree with the Reviewer. We have modified the claim in the manuscript to reflect only the demonstrated feedback response relative to a commercial DFB laser as has been done in other feedback insensitive laser papers (see below for more detail) and have moved calculation of the optical feedback parameter to the Supplemental. Additionally, we have adjusted the feedback plots (now Fig. 4) to report the optical feedback level as a ratio $\frac{P_{feedback}}{P_{out}}$ in dB rather than as an absolute power level in dBm to be consistent with other relevant feedback experiments [1-3]. The highest level of optical feedback in our experiment was -10 dB, limited primarily by the fiber-to-chip coupling and the connectors and splitters in the feedback loop. It is important to note that the optical feedback experiment we perform is nearly identical to the one reported in [1] (which was referenced by the Reviewer) where the frequency noise (FN) of the laser does not degrade even at the highest feedback level. Refs [1,2] measure that relative to a commercial DFB laser, their laser reaches coherent collapse at a feedback level of -41 dB. Using the methodology reported in Refs [1,2] we have demonstrated a 31 dB improvement in resilience to optical feedback compared to a commercial DFB. Ref [1] makes a similar claim, reporting that their highest feedback level of -6.9 dB without degradation in the FN is a demonstration of “over-34-dB improvement in the feedback insensitivity... equivalent to the effective isolation that optical isolators can provide.” We emphasize that our experiment is the first such demonstration with a Vernier-style tunable laser.

Fig. 1. Comparison of optical feedback experiments. (a, b) is the experimental setup and frequency noise plots from ref [1] and (c, d) are from our ECTL manuscript.

[1, 54] C. Xiang, W. Jin, O. Terra, B. Dong, H. Wang, L. Wu, J. Guo, T. J. Morin, E. Hughes, J. Peters, Q.-X. Ji, A. Feshali, M. Paniccia, K. J. Vahala, and J. E. Bowers, "3D integration enables ultralow-noise isolator-free lasers in silicon photonics," *Nature* **620**(7972), 78–85 (2023).
 [2, 43] M. Harfouche, D. Kim, H. Wang, C. T. Santis, Z. Zhang, H. Chen, N. Satyan, G. Rakuljic, and A. Yariv, "Kicking the habit/semiconductor lasers without isolators," *Opt. Express* **28**(24), 36466 (2020).
 [3, 49] Z. Zhang, K. Zou, H. Wang, P. Liao, N. Satyan, G. Rakuljic, A. E. Willner, and A. Yariv, "High-Speed Coherent Optical Communication With Isolator-Free Heterogeneous Si/III-V Lasers," *Journal of Lightwave Technology* **38**(23), 6584–6590 (2020).

To address Reviewer #1s comments, we have made the following changes to the manuscript:

From the Abstract:

“The hybrid integrated silicon nitride external cavity tunable laser is stabilized to a silicon nitride 10-meter long integrated coil-resonator without the need for an optical isolator by leveraging *the inherent 45 dB isolation.*”

→ “The hybrid integrated silicon nitride external cavity tunable laser is stabilized to a silicon nitride 10-meter long integrated coil-resonator without the need for an optical isolator by leveraging **the laser resilience to optical feedback of up to 30 dB beyond that of a commercial DFB laser.**”

From the Introduction:

“We demonstrate that the feedback resilience of the 3.5 million intrinsic-Q Vernier rings in the ECTL provide an inherent isolation of ~45 dB relative to a typical commercial III-V DFB laser.”

→ “We demonstrate that the feedback resilience of the 3.5 million intrinsic-Q Vernier rings in the ECTL provide an inherent isolation of ~30 dB relative to a typical commercial III-V DFB laser.”

From the Results:

Calculation of the optical feedback parameter has been moved to the Supplemental, and the experimental setup moved into Fig. 4. The following has been added to the manuscript.

The ECTL resilience to optical feedback is demonstrated by operating the PDH lock to the 10-meter coil reference cavity without an isolator – marking the first such demonstration for a Vernier-style laser and a key step toward fully integrated stabilized lasers on-chip. To further investigate this under controlled conditions, we utilized an optical feedback measurement setup⁴³⁻⁴⁵ (see Fig. 4a). Using a fiber circulator and a variable optical attenuator (VOA) we precisely control the feedback level back to the laser while accounting for circuit losses to determine the power reaching the laser. A small fraction of the ECTL output is diverted to measure the power and the frequency noise of the laser, the results of which are plotted in Fig. 4b. Compared to a commercial III-V DFB laser that experiences FN degradation and coherence collapse at a feedback level of -40 dB⁴³⁻⁴⁵, the ECTL maintains single-mode operation across all tested feedback levels, up to -10 dB, with no degradation in the frequency noise. This represents a 30 dB improvement in resilience to optical feedback over a commercial DFB laser and enables the isolator-free coil resonator stabilization.

We observe in Fig. 4b that the measured frequency noise of the ECTL decreases with increasing optical feedback. This is likely due to the long coherence time and strong mode selectivity of the ECTL where photons that re-enter from the external feedback loop remain phase-coherent with the intracavity field, effectively increasing the intracavity photon number and reducing quantum noise. Additionally, the extended fiber feedback circuit increases the overall optical mode volume, which suppresses TRN, leading to a lower frequency noise floor.

From the Discussion:

“isolation of 45 dB” → “**isolator free operation in a Vernier-style hybrid laser**”

Table 1 now records the optical isolation as **-30*** dB**

***** Relative to a typical commercial III-V DFB laser**

From the Methods:

The section on calculating the optical feedback parameter has been moved to the Supplemental, and replaced with further details on the experimental setup not discussed in the Results section.

Optical feedback measurements. The optical feedback measurement, illustrated in Fig. 4a, utilizes a fiber circulator and a variable optical attenuator (VOA) to control the optical feedback level. The power returning to the PIC is polarization sensitive and must be adjusted to maximize the optical feedback before making any measurements. The power returned to the ECTL is monitored by tapping 5% of the feedback power

and measuring on a power meter. Another small fraction of the laser power is diverted to an EDFA and then to an OFD setup to measure the frequency noise of the laser under different feedback conditions. The full fiber feedback circuit is ~25 meters long. All losses in the feedback circuit must be accounted for to most accurately determine the actual feedback power to the laser. The highest level of optical feedback in our experiment was -10 dB, limited primarily by the optical losses due to fiber-to-chip coupling and the connectors and splitters in the feedback loop.

The measurements in Fig. 5 suggest that self-injection locking (SIL) between the laser and the external fiber loop feedback circuit takes place. As SIL implies a frequency pulling effect on the laser, such optical feedback may hence affect the frequency stability of the laser even well below the level required to induce laser instability and chaos. Do the authors observe any frequency pulling, noise reduction or self-injection locking of the laser to the ring cavity when the two chips are connected? Injection locking of a kHz laser to an ultra-low noise cavity was demonstrated in Ref 44 and may also work in the case here.

We do not observe any frequency pulling associated with self-injection locking between the ECTL and the spiral ring cavity. We have experience with SIL (see Ref [1] below) and did attempt to SIL the laser to the coil in our experiment reported here, but did not observe any effect. Additionally, in doing the PDH lock we sweep the ECTL wavelength across the coil resonance to generate the PDH error signal, and see no evidence of intermittent frequency pulling in the locked-ECTL frequency noise plots. This is likely, in part, due to the fact that the level of back reflection that reaches the laser is quite low, due both to the fiber-chip coupling losses and the front mirror of the ECTL being ~ 75% reflective.

[1] A. Isichenko, A. S. Hunter, D. Bose, N. Chauhan, M. Song, K. Liu, M. W. Harrington, and D. J. Blumenthal, "Sub-Hz fundamental, sub-kHz integral linewidth self-injection locked 780 nm hybrid integrated laser," *Sci Rep* **14**(1), 27015 (2024).

What is the modulation frequency of the EOM for PHD locking?

10 - 20 MHz. We have added this information to the Results section of the manuscript:

We stabilize the ECTL to the 10-meter-long silicon nitride integrated coil reference cavity without an optical isolator between the laser and reference cavity using an EOM to generate locking sidebands of 10 - 20 MHz...

Reviewer #2 (Remarks to the Author):

The authors have pushed the state of the art by combining the above two techniques, enabling the external cavity laser to be locked to the coil cavity across various resonance within the full tuning range. The proposed laser is promising for many applications such as ultra-precise metrology, microwave photonics, and quantum optics. Given the record-breaking performance, solid experimental results and future potential impact, I recommend this paper to be accepted by Nature Communications after addressing the following issue.

1. My main concern is about the design and performance of external cavity laser. I understand that the authors employ large rings for sufficient linewidth reduction, by it may be difficult to support single-mode lasing. Based on the parameters given in the manuscript, I simulated the reflection spectrum of the Vernier-based external cavity by simply multiplying the transmissions of the two rings. The simulated adjacent sidelobe suppression of the external cavity is less than 0.5 dB. I am concerned that such a low level of sidelobe suppression may not be sufficient to ensure stable single-mode lasing, making mode hopping likely to occur. I hope the authors provide simulation results of the Vernier-based external cavity and address this issue.

We thank the Reviewer for this comment and appreciate that they took the time to simulate the transmission spectrum of the rings. We agree with the results of the Reviewer's simulation with regards to the Vernier reflection spectrum when considering only the two rings. We note however that the side mode suppression (SMSR) of the linear Vernier transfer function does not dictate the SMSR of the laser above threshold where nonlinear feedback must also be considered. A complete analytical description of the SMSR of the ECTL would require incorporating the wavelength dependence of the Sagnac loop mirror, the differential gain of the RSOA, the Fabry-Perot modes of the hybrid-integrated laser cavity, as well as the two-ring transmission spectrum. We also point out that in previous results the Vernier reflection spectrum SMSR is not indicative of the SMSR of the laser, for example in Refs [1, 2, 3, 4] the Vernier SMSRs range from 22, 20, 5 and 8.5 dB, respectively, and the measured laser SMSRs are as high as 64, 60, 68, and 75 dB, respectively. Furthermore, the results demonstrated in this manuscript, namely the ability to stabilize the ECTL to a high-Q cavity over long enough timescales to make frequency noise measurements using two independent methods, require stable single-mode operation -- any mode hopping ruins the PDH lock -- and therefore we believe represents demonstration of the laser stability that eases the Reviewer's concern.

- [1] Y. Wu, S. Shao, L. Tang, S. Yang, H. Chen, and M. Chen, "Hybrid integrated tunable external cavity laser with sub-10 Hz intrinsic linewidth," *APL Photonics* **9**(2), 021302 (2024).
 [2] P. A. Morton, C. Xiang, J. B. Khurgin, C. D. Morton, M. Tran, J. Peters, J. Guo, M. J. Morton, and J. E. Bowers, "Integrated Coherent Tunable Laser (ICTL) With Ultra-Wideband Wavelength Tuning and Sub-100 Hz Lorentzian Linewidth," *Journal of Lightwave Technology* **40**(6), 1802–1809 (2022).
 [3] Y. Wu, S. Shao, L. Tang, S. Yang, H. Chen, and M. Chen, "Hybrid integrated tunable external cavity laser with sub-10 Hz intrinsic linewidth," *APL Photonics* **9**(2), 021302 (2024).
 [4] Y. Guo, R. Zhao, G. Zhou, L. Lu, A. Stroganov, M. S. Nisar, J. Chen, and L. Zhou, "Thermally Tuned High-Performance III-V/Si₃N₄ External Cavity Laser," *IEEE Photonics Journal* **13**(2), 1–13 (2021).

2. The output power below 1 mW is significantly lower than over-10-mW power in other external cavity lasers. Please give an explanation and potential solutions, if any.

We agree that the output power is an important characteristic of laser performance and the Reviewer is right that specifically over-10-mW output power may be necessary for many applications. We measure off-chip output power of between 0.23 mW at 1520 nm and 4.37 mW at 1578 nm that includes a roughly 3 dB insertion loss from the PIC-to-fiber coupling. The wavelength dependence of the output power is predominantly attributed to the wavelength dependence of the Sagnac loop mirror. We also note that other Vernier-style hybrid lasers listed in Table 1 report similar or lower output powers, including Refs [1 – 3].

The output power is predominantly a function of the RSOA gain, the cavity losses and the loop back mirror reflectivity. Several explanations and potential solutions for the cavity loss include (1) the RSOA-to-PIC coupling loss, which can be improved by optimizing the SiN waveguide taper and by packaging (2) propagation loss which we estimate to be ~ 7.5 dB/m but have shown in previous work can be improved to be as low as 0.034 dB/m (3) ring-bus coupling optimized to reduce losses at the thru ports. Additionally, the reflectivity of the loop mirror directly relates to the output power. We estimate that at 1550 nm the reflectivity of the loop mirror is ~ 75%; lowering this reflectivity would serve to boost the output power of our device. We also mentioned in the Discussion section of the original manuscript, hybrid integration with multiple gain blocks has also been demonstrated with silicon nitride-based external cavity lasers and is a viable method to achieve much higher output power without compromising the narrow linewidth performance, see Ref. [4].

We have added the following to the Discussion section:

For applications that require high optical output power additional gain blocks can be added and operated in parallel with a shared high-Q silicon nitride external cavity²⁵ where output powers >100 mW have

already been demonstrated in a dual-gain hybrid-integrated laser⁴⁶. Other pathways to increasing the output power are to incorporate on-chip amplifiers^{55,56} or through injection-locked amplification.

- [1] Y. Wu, S. Shao, L. Tang, S. Yang, H. Chen, and M. Chen, "Hybrid integrated tunable external cavity laser with sub-10 Hz intrinsic linewidth," *APL Photonics* **9**(2), 021302 (2024).
 [2] P. A. Morton, C. Xiang, J. B. Khurgin, C. D. Morton, M. Tran, J. Peters, J. Guo, M. J. Morton, and J. E. Bowers, "Integrated Coherent Tunable Laser (ICTL) With Ultra-Wideband Wavelength Tuning and Sub-100 Hz Lorentzian Linewidth," *Journal of Lightwave Technology* **40**(6), 1802–1809 (2022).
 [3] Tran, M. A. et al. Ring-Resonator Based Widely-Tunable Narrow-Linewidth Si/InP Integrated Lasers. *IEEE Journal of Selected Topics in Quantum Electronics* **26**, 1–14 (2020).
 [4] Boller, K.-J. et al. Hybrid Integrated Semiconductor Lasers with Silicon Nitride Feedback Circuits. *Photonics* **7**, 4 (2020).

3. As shown in Fig. 2(d), lasing at a shorter wavelength requires a high heating power on the ring, subsequently leading to a higher thermorefractive noise. However, in Fig. 2(d), the fundamental linewidth is decreased with the shorter wavelength. Please explain this issue.

We appreciate the Reviewer’s important question. The thermorefractive noise, though dependent on the specific mode profile, does scale linearly with ambient temperature as the Reviewer notes and may cause the TRN of the rings to increase with higher heating power at shorter wavelengths; however two other effects may serve to offset and even reverse this: (1) the reflectivity of the loop mirror increases at shorter wavelengths due to the wavelength dependence of the evanescent coupler resulting in higher intracavity power (2) the ring-bus coupling strength decreases at shorter wavelengths which leads to longer photon lifetimes because the effective cavity length of the ring resonator increases with decreasing κ :

$$L_{eff} = \left(\frac{1}{2} + \frac{1 - \kappa}{\kappa} \right) L_{ring}$$

Both effects contribute to lower fundamental linewidths at shorter wavelengths, consistent with the measurements in Fig. 2(d).

We have added the following to Fig. 2(f):

Higher reflectivity of the Sagnac loop mirror and weaker ring-bus coupling may contribute to the decrease in FLW at shorter wavelengths.

4. In laser noise measurement, the authors give the estimated TRN and PTN curve in Supplementary Fig. 3, which fit the measured laser frequency noise curve well. Could the authors give more details about the calculation method and the corresponding estimation parameters, especially the PTN calculation that is rarely mentioned in the previous papers. Moreover, I suggest them to add this PTN curve in Fig. 2(e) in the manuscript.

We agree and per the Reviewer’s suggestion we have added the PTN and PD noise curves to Fig. 2(e) in the manuscript. The photo-thermal frequency noise spectrum induced by optical power fluctuations in the external cavity laser is described by,

$$S_{PTN}(f) = \left[\frac{\Delta f_{opt}}{P_{opt}} \right]^2 H_{th}^2(f) P_{opt}^2 S_{RIN}$$

where S_{RIN} is the experimentally measured relative intensity noise (RIN) spectrum (plotted in Supplemental Fig. 4(c)), P_{opt} is the estimated on-chip optical power, $H_{th}(f)$ is the thermal frequency response which can be estimated from COMSOL simulations, and $\frac{\Delta f_{opt}}{P_{opt}}$ describes the photo-thermal red shift strength that includes an estimation of ξ the absorption loss fraction which can be estimated experimentally. Our previous papers provide extensive details on this calculation, for example in Ref [1].

[1] K. Liu, *et al.*..., "Photonic circuits for laser stabilization with integrated ultra-high Q and Brillouin laser resonators," *APL Photonics* 7(9), 096104 (2022)

5. In the laser stabilization demonstration, the PDH error signal is directly fed into the RSOA. On account of the high-Q property of the external cavity, could this minor phase shift lead to large laser power fluctuation?

The PDH error signal feedback does affect the power fluctuations of the laser; where PTN dominates the frequency noise (FN) of the free running laser, the PDH feedback must suppress this RIN-induced PTN... in this case we can say that the PDH error signal leads to power fluctuations on the same order of magnitude as the original RIN of the free running laser. At higher offset frequencies there may be some additional power fluctuation (RIN) added by the PDH signal beyond the bandwidth of photothermal response and thus does not show up in the FN spectrum, but this would need to be measured experimentally by taking the RIN of the stabilized laser. We have not characterized the RIN of the stabilized laser here.

In addition, does the long photo lifetime of the external cavity laser limit the PDH locking bandwidth? Please comment on these issues, and a minor personal suggestion is to add the PDH residual noise curve in Fig. 4 (a).

In this experiment the PDH lock is limited primarily by the servo (electronics) bandwidth (~ 0.5 MHz). This is consistent with the locking bandwidth demonstrated in our other published stabilized laser results using a variety of lasers and reference cavities, see [1 – 3]. Unfortunately we did not record the PDH residual noise at the time of measurement.

[1] K. Liu, *et al.*..., "36 Hz integral linewidth laser based on a photonic integrated 4.0 m coil resonator," *Optica* 9(7), 770 (2022).

[2] S. Sun, *et al.*..., "Integrated optical frequency division for microwave and mmWave generation," *Nature* 627(8004), 540–545 (2024).

[3] K. Liu, *et al.*..., "Photonic circuits for laser stabilization with integrated ultra-high Q and Brillouin laser resonators," *APL Photonics* 7(9), 096104 (2022)

6. The laser feedback measurement results are strange. Commonly the laser performance deteriorates with the increased feedback light power, but the experiment shows the opposite result. As the authors claimed, it is possible that the laser enters a self-injection locking regime under specific phase-matching conditions. However, I think the experiment results do not validate the feedback insensitivity of the laser. Such a phenomenon should be avoided in the experiment, because the feedback light with random phase disrupts the oscillation in the practical scenarios. Moreover, Sidemode suppression ratio is a critical indicator in feedback measurement that should be characterized. I suggest the authors to repeat this experiment and refer to the methodologies outlined in two papers listed below. [1] Xiang, Chao, *et al.* "3D integration enables ultralow-noise isolator-free lasers in silicon photonics." *Nature* 620.7972 (2023): 78-85. [2] Tang, Liwei, *et al.* "A method for improving reflection tolerance of laser source in hybrid photonic packaged micro-system." *IEEE Photonics Technology Letters* 33.9 (2021): 465-468.

We thank the Reviewer for their feedback. As noted in the response above to Reviewer #1 the feedback measurement we performed is nearly identical to the one referenced by the Reviewer in Ref [1]. We describe above that we have modified our claim to be more consistent with the calculation in [1] that compares the insensitivity to a commercial DFB laser, which we now do. We include in our response to Reviewer #1 a side-by-side comparison. We note that they are also both similar to the one reported in Ref [3] where a fiber

feedback loop utilizing a circulator feeds the laser light back to itself, and the frequency noise is measured using an MZI as an optical frequency discriminator.

Setup from Ref [3] the Yariv “Kicking the Habit” paper:

[Figure Redacted]

The Reviewer also references Ref [2, Tang, et al.] that similarly utilizes a fiber feedback loop with a VOA, however instead of measuring the frequency noise they monitor the onset of coherent collapse using an optical spectrum analyzer.

[Figure Redacted]

As in Ref [1] we do not observe coherent collapse with our laser even at the highest feedback level of -10 dB, evidenced by the absence of discontinuities in the FN spectra measured using a 1 MHz FSR optical frequency discriminator. Additionally, we did take an OSA trace at a couple of the feedback levels, though unfortunately not at each wavelength, and measure SMSRs of ~ 57 dB. Since most of the output power of the laser is dedicated to the feedback loop an EDFA is necessary to measure the spectrum on an OSA. We have added these OSA traces to the ECTL feedback measurements figure, Fig. 4, in the manuscript.

Since the methodology we followed is nearly identical to the two that the Reviewer requests that we follow, we believe the measurement results should be acceptable. As mentioned in the response to Reviewer #1 we

revised the isolation claim (ex. resilience to optical feedback of up to 30 dB greater than a commercial DFB laser) to be consistent with how it is reported in Refs [1-3] and based on the highest optical feedback level measured in the experiment, rather than estimated from the feedback parameter.

[1] C. Xiang, W. Jin, O. Terra, B. Dong, H. Wang, L. Wu, J. Guo, T. J. Morin, E. Hughes, J. Peters, Q.-X. Ji, A. Feshali, M. Paniccia, K. J. Vahala, and J. E. Bowers, "3D integration enables ultralow-noise isolator-free lasers in silicon photonics," *Nature* **620**(7972), 78–85 (2023).

[2] Tang, Liwei, et al. "A method for improving reflection tolerance of laser source in hybrid photonic packaged micro-system." *IEEE Photonics Technology Letters* 33.9 (2021): 465-468.

[3] M. Harfouche, D. Kim, H. Wang, C. T. Santis, Z. Zhang, H. Chen, N. Satyan, G. Rakuljic, and A. Yariv, "Kicking the habit/semiconductor lasers without isolators," *Opt. Express* **28**(24), 36466 (2020).

1. The authors have done a good job in comprehensively comparing their work with the previous results, and Fig. 6 is very clear and valuable. Nevertheless, it is inappropriate to limit the comparison to hybrid integrated lasers while excluding some high-performance heterogenous integrated lasers (references are listed below). I recommend that the authors expand their comparison to include chip-scale lasers, particularly by adding Ref. [3], which demonstrates a state-of-the-art integral linewidth of 1 Hz. [3] Guo, Joel, et al. "Chip-based laser with 1-hertz integrated linewidth." *Science advances* 8.43 (2022): eabp9006. [4] Morton, Paul A., et al. "Integrated coherent tunable laser (ICTL) with ultra-wideband wavelength tuning and sub-100 Hz Lorentzian linewidth." *Journal of Lightwave Technology* 40.6 (2022): 1802-1809.

We thank the Reviewer for their feedback.

[4] is already included in Table 1 and Fig. 6 in the original document (listed as Ref [58]).

[3] was left out because although the publication title says "chip-based" it utilizes a bulk optic reference cavity to achieve the reported 1 Hz integral linewidth. We could have referenced the spiral cavity SIL results from [3], however, the authors published even better SIL results with the *same* 135 MHz spiral cavity in Li, B. *et al.* "Reaching fiber-laser coherence in integrated photonics". *Opt. Lett., OL* 46, 5201–5204 (2021). This reference is already included in Table 1 and in Fig. 6 of the original manuscript, listed as Ref [54].

Reviewer #3 (Remarks to the Author):

The manuscript presents a hybrid-integrated external cavity tunable laser (ECTL) stabilized to an integrated silicon nitride coil resonator. The authors claim record-low fundamental linewidths and improved noise performance without the need for an optical isolator. The paper highlights the potential for compact and robust photonic integration in ultra-narrow linewidth lasers for precision applications. My concerns are the following:

- The work primarily builds on previously published results on external cavity lasers and hybrid integration techniques. While the authors claim significant improvements in linewidth and stability, the advances are incremental rather than groundbreaking.

We thank the Reviewer for their feedback but disagree with their conclusion. No other hybrid integrated laser with an integrated reference cavity has demonstrated such low intrinsic and integral linewidths, and certainly not in a way that presents a path towards a stabilized laser on a single chip. We believe this is made clear in Table 1 and Fig. 6 in the original manuscript.

- The isolation effect is primarily attributed to the high-Q nature of the optical rings and the low-loss SiN platform by the same group has already been reported. Repurposing the high-Q ring resonators for a different application does not constitute a novel innovation and does not justify the claim of a significant advancement.

We strongly disagree with the Reviewer. We believe new applications of a core technology are perfectly acceptable grounds for publication. We have not published a hybrid-integrated Vernier laser before nor have we demonstrated PDH locking of such a laser to an integrated coil reference cavity, and also both chips fabricated using the same silicon nitride thickness and fabrication process for both. To the best of our knowledge this is also the first demonstration of resilience to optical feedback for a Vernier-style hybrid laser.

- What is the reason for quite high-frequency noise at lower frequency offsets?

We have measured down to 1 Hz offset which is quite low, and one would expect the noise to be higher at closer frequencies due to environmental and other technical noise sources. Other comparable publications only report down to 1 kHz so our low frequency noise level will look higher, but this is not the case for the 1 – 1 kHz range. We refer the Reviewer to Fig. 6 to point out that the frequency noise of the free-running laser at low frequency offsets is comparable to or lower than most of the other published results, and the FN of the stabilized laser is lower than other previously published hybrid-integrated results. For the free running laser, photothermal noise (PTN) dominates at lower frequency offsets and is plotted in Supplemental Fig. 3. We have added estimates for PTN and photodetector noise from Supplemental Fig. 3 to the main manuscript Fig. 2(e) per the request of Reviewer #2. For the stabilized laser the ECTL inherits the frequency stability of the 10-m coil resonator, so the frequency noise at low frequency offsets is dominated by environmental noise (eg. thermal, vibrational).

- The EOM and spiral cavity are not integrated, which could have added to the novelty of the manuscript.

We agree that incorporating an integrated sideband modulator and adding the coil cavity to the same chip as the ECTL would add to the novelty of the result and is a focus of future work. We remark on this in the Discussion section, and the results in this paper show that it should be possible, and without the need for an optical isolator, which otherwise, to-date, would make further integration prohibitive. We also note two design approaches that eliminate the need for an EOM or AOM between the ECTL and spiral resonator:

the EOM used in this experiment can be eliminated by adding a PZT-on-Si₃N₄ integrated double sideband modulator that can achieve locking bandwidths up to 20 MHz [1], or by employing modulation free stabilization techniques [2,3]. Thermal and PZT actuators have been demonstrated in this Si₃N₄ platform without affecting waveguide loss, independent of wavelength, and operate from DC out to 10s of MHz [4] and could also be used to directly tune the intracavity ECTL rings rather than feeding the PDH error signal back to the gain chip current.

[1] Wang, J., Liu, K., Rudy, R. Q. & Blumenthal, D. J. in *Optica Quantum 2.0 Conference and Exhibition*. QW3B.3 (Optica Publishing Group).

[2] Idjadi, M. H., Kim, K. & Fontaine, N. K. Modulation-free laser stabilization technique using integrated cavity-coupled Mach-Zehnder interferometer. *Nat Commun* **15**, 1922 (2024).

[3] Liu, K., Harrington, M. W., Wang, J., Nelson, K. D. & Blumenthal, D. J. in *Frontiers in Optics + Laser Science 2023 (FiO, LS)*. JW4A.22 (Optica Publishing Group).

[4] Wang, J., Liu, K., Harrington, M. W., Rudy, R. Q. & Blumenthal, D. J. Silicon nitride stress-optic microresonator modulator for optical control applications. *Opt. Express* **30**, 31816-31827 (2022).

- The impact of the paper is very similar to Guo et al., Science Advances (2022) (<https://www.science.org/doi/10.1126/sciadv.abp9006>) with little differentiation.

We strongly disagree with the Reviewer's assertion that this work is *very similar* to [1, Guo, *et al.*... Science Advances (2022)] for several reasons: (1) Guo, *et al.*'s injection-locked laser does not feature wide tuning (< 1 nm), whereas our Vernier-style laser tunes across 60nm. In another paper where the same group published a separate result [2] with the *same* 135 MHz spiral cavity they state that the SIL laser tunes across only 0.8 nm. This is not sufficient for many applications. (2) Guo, *et al.*... PDH lock the SIL spiral laser to a bulk optic reference cavity, *not* an integrated cavity. (3) Guo, *et al.* utilize an isolator between their SIL spiral cavity laser and the bulk optic component, making any further progress towards a genuine "chip-scale" stabilized laser nontrivial. These are significant differences between our work and Guo, *et al.*... that should not be discounted.

[1] J. Guo, C. A. McLemore, C. Xiang, D. Lee, L. Wu, W. Jin, M. Kelleher, N. Jin, D. Mason, L. Chang, A. Feshali, M. Paniccia, P. T. Rakich, K. J. Vahala, S. A. Diddams, F. Quinlan, and J. E. Bowers, "Chip-based laser with 1-hertz integrated linewidth," Science Advances **8**(43), eabp9006 (2022).

[2] B. Li, W. Jin, L. Wu, L. Chang, H. Wang, B. Shen, Z. Yuan, A. Feshali, M. Paniccia, K. J. Vahala, and J. E. Bowers, "Reaching fiber-laser coherence in integrated photonics," Opt. Lett., OL **46**(20), 5201–5204 (2021).

- The manuscript claims that the ECTL inherently provides 45 dB of isolation. However, this assertion is not well supported by experimental comparisons with commercially available isolators or experimental data.

We thank the Reviewer for their feedback and have revised the isolation claim to be compatible with the language used in Ref [1]. As described in detail in response to Reviewer #1 we have adjusted the manuscript to revise the isolation claim: rather than asserting 45 dB of inherent isolation based on an estimate of the optical feedback parameter we instead claim **resilience to optical feedback of up to 30 dB beyond that of a commercial DFB laser**. Where this claim is supported by the optical feedback experiment that shows no degradation of the frequency noise of the laser with feedback of up to -10 dB, and consistent with the methodology and results reported in Refs [1] and [2].

[1] C. Xiang, W. Jin, O. Terra, B. Dong, H. Wang, L. Wu, J. Guo, T. J. Morin, E. Hughes, J. Peters, Q.-X. Ji, A. Feshali, M. Paniccia, K. J. Vahala, and J. E. Bowers, "3D integration enables ultralow-noise isolator-free lasers in silicon photonics," Nature **620**(7972), 78–85 (2023).

[2] M. Harfouche, D. Kim, H. Wang, C. T. Santis, Z. Zhang, H. Chen, N. Satyan, G. Rakuljic, and A. Yariv, "Kicking the habit/semiconductor lasers without isolators," Opt. Express **28**(24), 36466 (2020).

Given the incremental nature of the work and lack of novelty, I do not recommend this manuscript for publication in Nature Communications.

We strongly disagree with the Reviewer that this work is merely incremental and lacks novelty for all the reasons outlined in detail above and in the revised manuscript. We believe this point is belied by Table 1 and Fig. 6 from the original manuscript, which the Reviewer does not directly address. We report record low fundamental and integral linewidth and across a 60 nm tuning range. We also demonstrate this with isolator-free operation - a first for a Vernier-style laser – and in a platform that enables full integration.

Appeal Response to Nature Communications

“Ultra-low linewidth coil stabilized widely tunable isolator-free hybrid integrated external cavity laser”

Manuscript NCOMMS-24-78214

We would like to thank the reviewers for their valuable time, questions, feedback, and comments. The points raised by the reviewers and their active involvement have resulted in a greatly improved manuscript as well as more effective communication of the significance of these results to a broad audience.

We have provided below a detailed response on a point-by-point basis for each reviewer’s comment (*blue*) with our response (*black*) and an excerpt of the updated text (*red*).

Reviewer #1 (Remarks to the Author):

D.A.S. Heim et al. have cured my two concerns with the manuscript relating to figures 1 and 3 and the claim of 45 dB isolation claim. From a technical point of view, the manuscript could be published. Whether the novelty of the work merits publication in Nature Communications is left to the editors.

We thank Reviewer #1 for their comments and helping make this a better manuscript. Thank you for the feedback that your main concerns have been addressed and for the recommendation to publish from a technical perspective.

Reviewer #2 (Remarks to the Author):

I think the revised manuscript has adequately addressed all concerns raised during the review process and is now acceptable for publication.

We thank Reviewer #2 for their time and feedback that helped make this an improved manuscript. We also want to thank Reviewer #2 for the recommendation to publish our work.

Reviewer #3 (Remarks to the Author):

I appreciate the authors’ detailed rebuttal and clarifications regarding the technical contributions and distinctions from prior work. There is no doubt that the manuscript reports a high-performance hybrid integrated ECTL with impressive metrics, particularly in achieving a record-low integral linewidth and in demonstrating potential pathways toward a fully stabilized on-chip laser.

We thank Reviewer #3 for their detailed feedback and time spent re-reviewing our manuscript, which has resulted in an improved manuscript. We also want to thank Reviewer #3 for recognition that we report record results.

However, I maintain that while the laser's performance represents an incremental advancement, the core novelty of a fully integrated on-chip stabilized laser remains a future goal rather than a realized achievement in this work. As the authors themselves acknowledge, the spiral resonator is not integrated on the same chip as the ECTL, and the system still relies on discrete components such as the external electro-optic modulator (EOM). While the authors suggest feasible paths for future integration, such as using PZT-on-Si₃N₄ or modulation-free stabilization techniques, these remain prospective developments.

We respectfully disagree. Our results pave a direct path towards realizing the goal of a fully integrated stabilized laser on a chip. The critical steps towards achieving that goal are demonstrated here for the first time, including fabrication of the widely tunable laser and coil reference cavity using exactly the same 80-

nm thick silicon nitride low-loss platform, operation of the PDH lock without optical isolation, demonstrating of locking to the coil across a 40 nm tuning range with the performance of record low fundamental and integral linewidths across the whole tuning range. We believe these results mark a significant, rather than incremental, advance.

The comparison with Guo et al. is appreciated, and the authors correctly point out important distinctions in tuning range, reference cavity integration, and use of isolators. Nevertheless, it should be noted that their own spiral resonator is not co-integrated on the same photonic chip as the laser, and therefore, the claim of achieving a fully on-chip stabilized laser, which is central to the manuscript's novelty, is not yet substantiated in this implementation.

We would like to emphasize that we have not claimed a fully integrated stabilized widely tunable laser on a single chip. However, the technologies and performance we demonstrate have a direct path to doing so. This is what distinguishes our result from that published by Guo et al. 2022 in critically important ways and highlights the novelty of the work presented in the manuscript, for the reasons pointed out above by Reviewer #3. The work presented by Guo does not have a path to full integration due to the bulk optic reference cavity and the need for optical isolation, nor does the design reported by Guo support wide tunability.

The device architectures, and results are fundamentally different.

- Guo et al. 2022 utilizes an integrated EDBR laser that is self-injection locked to an integrated 135 MHz spiral resonator, which is then stabilized to a bulk-optic reference cavity.
 - o The self-injection locked EDBR fundamentally limits the device tuning range to < 1 nm. This is in contrast to our 60 nm tuning range.
 - o The frequency noise performance of the self-injection locked EDBR in Guo is not as low as what we report. See the discussion below.
 - o The reference cavity does not support tuning at a large number of tunable wavelengths, nor is this explored. Our coil supports locking on a high resolution 50 MHz grid over the whole laser 60 nm tuning range.
 - o The Guo paper does not demonstrate or explore the potential for isolator-free operation in the PDH lock to the bulk optic reference cavity. Ours provides a level of reflection tolerance (isolation) that enables direct connection of the laser to the coil reference cavity.
 - o There is no pathway to integrating the vacuum-gap bulk optic reference cavity and laser on chip. For our work the laser and coil resonator chips are fabricated in exactly the same waveguide platform with exactly the same fabrication process.
 - o The Guo paper does not explore frequency noise and linewidth over a large tuning range

Below, for the purpose of this response to Reviewer #3, we have plotted the frequency noise of Guo et al. In addition to the tuning range in Guo being negligible, the frequency noise and integral linewidths are below that reported for our coil-stabilized ECTL. Guo reports integral linewidth of 650 Hz, whereas we measure 27 Hz for the coil-locked ECTL, with nearly four orders of magnitude lower frequency noise between 10 – 100 Hz offset. We believe that demonstrating this level of performance in a hybrid-integrated tunable laser with a direct path towards full integration of the reference cavity represents a meaningful and novel result.

Additionally, the wide tunability achieved via Vernier tuning, although valuable, is not novel in itself, as this technique has been well demonstrated in previous literature across various platforms.

We agree with the Reviewer that the wide tunability of Vernier-style lasers has been reported. However, achieving such low fundamental and integral linewidths across a wide range of wavelengths in an integrated laser, as well as isolator-free operation, has not been demonstrated before and widens the potential application space for these types of lasers.

I believe the manuscript makes a valuable technical contribution and would be of interest to the photonics and laser stabilization community. However, given that the main novelty remains partially realized, I feel that the manuscript would be more appropriately positioned in a specialist journal with a focus on integrated photonics or laser engineering rather than in Nature Communications, which typically prioritizes transformative advancements with immediate and broad impact.

We thank the Reviewer for their feedback which has resulted in a more effective manuscript. We appreciate the Reviewer's recognition of the valuable technical contributions presented in this work and of its interest to the photonics community. At the same time we respectfully disagree and believe that this manuscript represents a transformative advancement in the potential of high-performance stabilized integrated laser technology that will be of immense interest to the Nature Communications readership.